# Technique, Feasibility, Utility, Limitations, and Future Perspectives of a New Technique of Applying Direct In-Scope Suction to Improve Outcomes of Retrograde Intrarenal Surgery for Stones

**DOI:** 10.3390/jcm11195710

**Published:** 2022-09-27

**Authors:** Vineet Gauhar, Bhaskar Kumar Somani, Chin Tiong Heng, Vishesh Gauhar, Ben Hall Chew, Kemal Sarica, Jeremy Yuen-Chun Teoh, Daniele Castellani, Mohammed Saleem, Olivier Traxer

**Affiliations:** 1Department of Urology, Ng Teng Fong General Hospital (NUHS), Singapore 609606, Singapore; 2Department of Urology, University Hospitals Southampton, NHS Trust, Southampton SO16 6YD, UK; 3Global Indian International School, Singapore 828649, Singapore; 4Department Urol Sci, University of British Columbia, Urologic Sciences, Vancouver, BC V6T 1Z4, Canada; 5Department of Urology, Biruni University Medical School, Istanbul 34010, Turkey; 6S.H.Ho Urology Centre, Department of Surgery, The Chinese University of Hong Kong, Hong Kong 96H2+Q9, China; 7Urology Unit, Azienda Ospedaliero-Universitaria Ospedali Riuniti di Ancona, Università Politecnica delle Marche, 60126 Ancona, Italy; 8Urology Research and Daycare Center, Apis Kidney Stone Institute, Mangalore 57502, India; 9Department of Urology AP-HP, Sorbonne University, Tenon Hospital, 75020 Paris, France

**Keywords:** renal stones, retrograde intrarenal surgery, laser lithotripsy, suction

## Abstract

Retrograde intrarenal surgery (RIRS) is accepted as a primary modality for the management of renal stones up to 2 cm. The limitations of RIRS in larger volume stones include limited visualization due to the snow-globe effect and persistence of fragments that cannot be removed. We describe a new, simple, cost-effective modification that can be attached to any flexible ureteroscope which allows simultaneous/alternating suction and aspiration during/after laser lithotripsy using the scope as a conduit to remove the fragments or dust from the pelvicalyceal system called direct in-scope suction (DISS) technique. Between September 2020 and September 2021, 30 patients with kidney stones underwent RIRS with the DISS technique. They were compared with 28 patients who underwent RIRS with a 11Fr/13Fr suction ureteral access sheaths (SUASs) in the same period. RIRS and laser lithotripsy were carried out traditionally with a Holmium laser for the SUAS group or a thulium fiber laser for the DISS group. There was no difference in age, gender, and history of renal lithiasis between the two groups. Ten (40%) patients had multiple stones in the DISS groups, whilst there were no patients with multiple stones in the SUAS group. Median stone size was significantly higher in the DISS group [22.0 (18.0–28.8) vs. 13.0 (11.8–15.0) millimeters, *p* < 0.001]. Median surgical time was significantly longer in the DISS group [80.0 (60.0–100) minutes] as compared to the SUAS group [47.5 (41.5–60.3) minutes, *p* < 0.001]. Hospital stay was significantly shorter in the DISS group [1.00 (0.667–1.00) vs. 1.00 (1.00–2.00) days, *p* = 0.02]. Postoperative complications were minor, and there was no significant difference between the two groups. The incidence of residual fragments did not significantly differ between the two groups [10 (33.3%) in the DISS group vs. 10 (35.7%) in the SUAS group, *p* = 0.99] but 10 (33.3%) patients required a further RIRS for residual fragments in the DISS group, whilst only one (3.6%) patient in the SUAS group required a subsequent shock wave lithotripsy treatment. Our audit study highlighted that RIRS with DISS technique was feasible with an acceptable rate of retreatment as compared to RIRS with SUAS.

## 1. Introduction

Retrograde intrarenal surgery (RIRS) has gained popularity and is accepted in guidelines as a primary modality for the management of renal stones up to 2 cm [1]. With the advent of newer-generation flexible ureterorenoscopes, Holmium:YAG laser lithotripsy, and Thulium fiber, this trend is also being exploited in stones larger than 2 cm [2], pediatric cases [3], anomalous kidneys [4] and competing with percutaneous nephrolithotripsy (PCNL) [5]. However, despite the advances in technology, there are challenges in achieving a high stone-free rate (SFR) in large stones greater than 2 cm with series reporting SFR between 56.7% (1) in single procedure to 89.3% in a mean average of 1.6 procedures [6] and in anomalous kidneys with SFR of 76.6% [7]. Hence, much consideration needs to be given to choose RIRS over PCNL as the first choice of intervention.

Despite the advantages of using MOSES technology and thulium fiber laser (TFL) which are faster, more efficient, and give fine dust [8,9], the limitations of RIRS in larger volume stones include limited visualization due to the snow-globe effect [10], the persistence of fragments that cannot be removed [11], and need for additional accessories such as baskets or maneuvers such as repeat surgery [12], table tilting techniques [13], percussion inversion diuresis [14], external physical vibration lithecbole (EPVL) [15] and the use of conventional and flexible suction ureteral access sheaths (SUAS) as well as steerable multi-lumen irrigation/aspiration devices to aspirate the fragments from different calyces [16,17]. All the above have variable outcomes either limited by the inability to reach the fragments, cost, lack of facilities, or know-how on how to use the technology at the place of practice.

Of the options available, the easiest and best way seems to be reaching the calyx wherein lie the pieces/dust and to directly aspirate the same. In the present study, we describe our experience with a simple, cost-effective innovation and modification that can be made by the surgeon and attached to any flexible or semi-rigid scope which allows simultaneous or alternating suction and aspiration during or after laser lithotripsy using the scope as a conduit to remove the fragments or dust in all parts of the pelvicalyceal system called direct in-scope suction (DISS) technique. We also compared the DISS technique with standard RIRS performed with a SUAS.

## 2. Materials and Methods

### 2.1. Audit Study

A retrospective analysis of prospectively collected data from 2 centers was performed between September 2020 and September 2021 to compare the feasibility, utility, and limitations of the DISS technique in real-life practice and to see if this procedure was reproducible, as well as observing the outcomes in relation to performing RIRS with the use of a SUAS. Any adult patient, symptomatic and asymptomatic, diagnosed with renal stones only, was counseled and consented to RIRS as a second stage after pre-stenting. There were no strict exclusion criteria except if the patient’s habitus was abnormal or she/he had an anatomical variation of his pelvicalyceal system. In the patients on whom a reusable scope was used, a strict post-scope assessment was made by the hospital’s biomedical department to test for any tip, working channel damage, or leak after each procedure and to report the same with photographic evidence. Patients received either of the procedures based on the surgeon’s own preference and were included only if ureteric access was feasible. Patients in whom a ureteric access for RIRS was impossible on the first attempt were then pre-stented and excluded from this study. The primary outcome was difference in SFR between the two techniques. The secondary outcome was difference in complications.

The study was conducted retrospectively following the 1964 Declaration of Helsinki and its later amendments. Data collection was obtained for clinical purposes, and all the procedures were performed as part of routine care. All patients signed an informed consent.

### 2.2. Direct In-Scope Suction: Assembling the Device

To facilitate the DISS technique, we took two 3-way stoppers and connected them. One inlet was used to connect suction tubing and the other for irrigation (Figure 1A). This completely versatile attachment can now be connected to any scope working inlet (Figure 1B). Next, the irrigation and suction tubes were attached to the inlet and outlet, respectively (Figure 1C,D; Appendix A).

The pressure-controlled Endoflow II^®^ single chamber pump (Rocamed, France) or gravity-based fluid control was used and the suction was connected either to a portable suction or directly to wall suction. Pressures were kept low throughout. This helped to facilitate adequate distention of the pelvicalyceal system during lithotripsy and active suctioning (Appendix A).

### 2.3. Surgical Techniques

Two consultant urologists performed all procedures. RIRS and laser lithotripsy were carried out traditionally with a Holmium 100 watt laser (Lumenis Pulse™ 100H, Boston Scientific, Marlborough, MA, USA) for the SUAS RIRS group or a TFL 35 watt (Urolase SP, IPG Photonics, Oxford, MA, USA) for the DISS technique group. A 7.5 Fr reusable ureteroscope (Flex-X2, Karl Stortz, Tuttlingen, Germany) or a single-use 7.5 Fr ureteroscope Uscope (Zhuhai Pusheng Medical Technology Co., Ltd., Zhuhai China) were used in the DISS group, whereas only a single-use 7.5 Fr ureteroscope Uscope, (Zhuhai Pusheng Medical Technology Co., Ltd., Zhuhai China) with a 11Fr/13Fr dual lumen SUAS ClearPetra™ (Wellead, Panyu, China) were used in the SUAS group.

Pop-dusting or pure dusting technique was used as needed in both groups [18]. We deployed continuous irrigation throughout, ensuring that the pelvicalyceal system was adequately drained by saline and to ensure intrarenal temperature was properly maintained, without compromising any principles of RIRS surgery [19]. The 3-way adapter for suction was sequentially opened either when a snow-globe effect was noted to aspirate the fragments or intermittently after lithotripsy to aspirate the dust [10]. The scope was targeted to the desired calyx where lithotripsy was performed and by retaining the scope in situ the dust was aspirated before proceeding with further lithotripsy. Post-lithotripsy, an inspection of all calyces was made, and by directing the tip of the scope and applying suction to the targeted area, all debris was aspirated including any inadvertent blood clots. This ensured adequate clearance by visual inspection. This procedure was satisfactorily done in all cases independent of the use of UAS and with all scopes in all cases. The presence of laser fiber did not interfere with the technique. Post-procedure, all patients had a double J stent placed as standard of care in both groups.

### 2.4. Patient Follow-Up

Postoperative follow-up was performed at 3 weeks to evaluate complications and assess residual fragments which were evaluated with KUB and ultrasound or computed tomography scan if deemed necessary. Stone-free status was defined as the absence of a single residual fragment >3 mm or in the absence of multiple fragments of any size.

### 2.5. Statistical Analysis

Continuous variables were presented as median and 25th–75th percentiles. Categorical variables were reported as absolute frequency and percentage. The Mann–Whitney U test was used to assess the difference between the two groups for continuous variables, whereas the Chi-square test was used for categorical variables. Statistical significance was set at a two-tailed *p*-value < 0.05. Statistical tests were conducted using SPSS software package version 25.0 (IBM Corp., Armonk, NY, USA).

## 3. Results

During the study period, 58 patients were included in the audit study. Thirty patients underwent DISS RIRS and 28 patients SUAS RIRS. Table 1 shows patients’ characteristics.

There was no difference in age, gender, and history of renal lithiasis between the two groups. Ten (40%) patients had multiple stones in the DISS groups, whilst there were no patients with multiple stones in the SUAS group. Median stone size was significantly higher in the DISS group [22.0 (18.0–28.8) vs. 13.0 (11.8–15.0) millimeters, *p* < 0.001]. A significantly higher number of patients had stones in the middle renal pole in the DISS group (24% vs. 11%, *p* < 0.001). Median surgical time was significantly longer in the DISS group [80.0 (60.0–100) minutes] as compared to the SUAS group [47.5 (41.5–60.3) minutes, *p* < 0.001]. There were no intraoperative complications in both groups. Hospital stay was significantly shorter in the DISS group [1.00 (0.667–1.00) vs. 1.00 (1.00–2.00) days, *p* = 0.02]. Postoperative complications were minor, and there was no significant difference between the two groups. The incidence of residual fragments did not significantly differ between the two groups [10 (33.3%) in the DISS group vs. 10 (35.7%) in the SUAS group, *p* = 0.99] but there was a greater number of patients with multiple residual fragments in the DISS group [8 (26.7%)] than in the SUAS group [5 (17.95)]. Ten (33.3%) patients required a further RIRS for residual fragments in the DISS group, whilst only one (3.6%) patient in the SUAS group required a subsequent shock wave lithotripsy treatment.

## 4. Discussion

In RIRS, a surgeon’s aim for successful laser lithotripsy is complete fragmentation of any stone in any location, avoiding complications, and total fragment removal to avoid clinically significant residual fragments, which can have a dubious course in follow-up [11]. Many studies report a diameter of 4 mm or 2 mm as clinically insignificant residual fragments [20]. Yet, as reported by Lovegrove et al., the overall risk of admission varied from 14% to 19%, and the risk of intervention from 12% to 35% for residual stones [21]. Further, there is no consensus whether basketing or dusting alone is a superior technique to render a patient stone-free post-RIRS, as reported by the EDGE research consortium [22]. Traditionally the focus in RIRS has been on scopes, lasers and utility of access sheaths, irrigation devices and the ability to effectively yet safely perform RIRS. Amongst various techniques deployed for improving RIRS outcomes a recent focus has been on using suction in RIRS. The utility of SUAS has been reported in several single-center studies [23,24]. Proposed advantages are that by adding suction, not only can we have lower intrarenal temperatures and pressure, but we can minimize infective complications, improve surgical times and improve SFR compared to traditional RIRS. Although the methodology is variably described in the literature, most papers cite that the limitation of using a SUAS is that fragments have to be brought to the sheath for effective removal [25]. This can be achieved by using baskets for larger fragments or active flushing of the pelvicalyceal system which can be time-consuming and needs assistance, and the use of accessories adds to the cost. The DISS technique was formulated to overcome these very limitations yet maintain the principles of traditional RIRS; where endourologists can continue to perform RIRS in the exact same traditional way with a simple add-on modification, as described in the surgical technique, to perform DISS. In our audit, all stones were accessed and laser lithotripsy was successfully performed for all stone sizes, and locations, even if 10 (33.3%) patients required a further RIRS. The technique did not pose any limitation to the procedure. Since we were using the scope as a suction conduit to aspirate, it was quintessential to use the pure dusting technique, hence TFL laser was preferentially used in all these cases, as it has been shown to be more effective and faster in dusting than the holmium laser [26,27]. In our series, patients with DISS had a longer operative time than SUAS, as even though laser lithotripsy was fast, it takes time to inspect each calyx and visually aspirate the dust and debris to ensure complete clearance. Further, the stone size was significantly larger in the DISS group along with a greater number of multiple stones in the DISS cohort, whereas the SUAS group had only solitary stones. The safety of the DISS technique was shown by the ability to have all patients discharged within 24 h of surgery. Although fever was reported in 11 cases, none of these patients had sepsis and were discharged with appropriate antibiotics and with zero readmission rate, a feature also noted in the SUAS group. While the DISS procedure is safe, we did note that despite our best attempt at the inspection of all calyces, 10 patients (33.7%) had residual fragments, similar to SUAS, with 26.7% noted to have multiple fragments. A limitation of our audit study was that the post-operative imaging carried out as part of an early follow-up study (3 weeks) in the DISS group was either an x-ray or CT scan and all the 10 patients with RF who had only had an X-ray were counseled for a repeat RIRS. However, all 28 patients who had SUAS underwent a CT scan, and based on the RF location and size only one patient needed ESWL as an ancillary procedure. This reiterates the need for proper imaging modality and timing of re-imaging after RIRS or any endourological intervention. Danivolic et al. (28) advise that if endoscopic evaluation post-procedure suggests that residual fragments are <2 mm, a 90th post-operative day non-contrast CT should be performed and KUB had only 31.6% (6/19) sensitivity to detect residual fragments >2 mm and did not add sensitivity or specificity to ultrasound.

We were able to successfully apply the DISS technique irrespective of the stone size location and numbers without any recorded damage to the scope. Our greatest limitation was the inability to conduct a randomized controlled study, as one of the centers did not have access to a suction UAS, a limitation which is seen in many centers, hindering the adoption of the technique. We duly note that this is a new procedure needing validation from further studies in more centers. We hope that the below-mentioned technique will help interested surgeons overcome teething issues to adopt this technique.

### 4.1. Intraoperative Procedural Tips, Tricks and Overcoming Limitations

We describe the operative technical challenges we faced during our learning curve and the recommendations on how to easily overcome them:

(1) Whilst the DISS technique is easy, it may take some experience for the endourologist to regulate the 3-way adaptors and perform laser lithotripsy simultaneously. We took about five cases before we were intuitively able to handle the regulators and this might also explain why ten patients required a further RIRS for their residual fragments.

(2) Application of suction both for DISS and using a SUAS requires an increased flow rate for irrigation than conventional RIRS as the fluid gets rapidly aspirated. We overcame this by using a 3-L saline bag and seldom needed a replacement.

(3) As suction is applied dust is aspirated along with blood clots, if any. This can pose a challenge in blocking the working channel with subsequent ineffective suction. This is easily overcome by intermittently flushing the scope with 10–20 mL of saline.

(4) The DISS technique relies on the ability to maneuver the tip of the scope to the desired area/calyx for aspiration and hence we suggest using a scope with a smaller tip. In our series both the Uscope and KarlStorz flex X2 were very suitable.

(5) Using the DISS technique gives complete flexibility to the urologist to work in any part of the kidney and can be used during lithotripsy or just after dusting. We observed that when applying DISS it was helpful to pause lithotripsy, aspirate the debris/dust and then proceed. This may slow the surgeon down but time spent usefully here can avoid an unnecessary repeat procedure. As with any procedure with experience, this is easily overcome.

(6) The key to a successful aspiration is to have fine dust. By Tenon definition [28], all particles ≤250 µm from all stone types were in agreement with all three criteria defining stone dust: (A) spontaneous floating under 40 cm H_2_O irrigation pressure; (B) mean sedimentation time of >2 s through 10 cm saline solution; (C) fully suitable for aspiration through a 3.6 F working channel.

This was particularly true for us when using the DISS technique and we recommend that surgeons should maximally dust the stone for ease of aspiration as fragments might not pass through most commercially available scope channels. However, this limitation is also overcome when any UAS is used as bigger fragments may be flushed out or basket out through the sheath.

(7) Large volumes of generated dust in big stones can create a snow-globe effect [10]. Using DISS and intermittently aspirating dust with a low power suction helps keep vision clear throughout the procedure; this has a sevenfold advantage:

(a) Prevents inadvertent injury to the mucosa of PCS.

(b) Exposes the hidden fragments under the dust which are better tackled by laser lithotripsy.

(c) Allows precise lithotripsy as the stone is visualized

(d) Helps tackle large volume stones with the same efficiency as smaller stones maintaining the safety and principles of RIRS procedure. An inference can be drawn as we did not have to abandon any DISS procedure and were able to tackle any stone in any location. There were no intra-operative complications reported. Potentially, if validated by larger multicenter studies this may indeed become the next milestone for managing larger stone volumes.

(e) Collected dust can be sent for analysis if the center is well equipped with this facility (Figure 2). Recently, Sierra Del Rio et al. have reported that by analyzing aspirated dust through the ureteroscope working channel by physical techniques, we can understand the lithogenic process of the urinary stone, without needing to analyze the stone fragment [29].

(f) Whilst we could not measure temperature and intrarenal pressure whilst using DISS, other reports have conclusively proven that adding suction potentiates this effect in RIRS, mitigating potential complications related to uncontrolled temperatures and pressures in the pelvicalyceal system, especially when using newer powerful lasers [16,17,19]. None of our patients in either cohort had pelvicalyceal injuries, life-threatening sepsis, or readmission for infectious-related complications.

(g) DISS technique could potentially minimize usage of accessories such as baskets for fragment removal and by suction aspiration of all the dust improve SFR. Yet, it is indeed safe to insert and manipulate the scope. In fact, the no-touch technique had been described as a safety standard of care by Grasso [30]. We experienced no difficulty and only for those patients in whom access was possible easily did we proceed to perform surgery.

(8) Surgeons using suction either by DISS or SUAS must be aware that mild pelvicalyceal system bleeding can be expected if too much suction pressure is applied which causes mild accidental mucosal petechial injury or inadvertent suctioning of the mucosa due to a collapsed system or due to rubbing of the sheath when it is moved across the ureteropelvic junction into the pelvicalyceal system to aspirate fragments. In our experience, this was not bothersome as it does not hamper vision, is transient and stops in a few seconds when suction is temporarily released and adequate irrigation is provided. However, caution must be exerted in being gentle with manipulation of the scope or UAS in the pelvicalyceal system.

### 4.2. Future Directions

(1)As for any procedure, we plan a randomized trial comparing DISS with traditional RIRS and vis a vis other commercially available SUAS.(2)Ergonomics plays a major role in RIRS and we plan to adopt the lessons learned in our first audit and improve on the design module for DISS [31,32].(3)The minimalist, simplistic, and cost-effective construction of the two 3-way stoppers is the cornerstone for a highly effective DISS technique. As the regulation of the two valves could be cumbersome to some, there remains potential for improving the design module while retaining its cost-effectiveness, functionality, and utility.(4)We need to conduct more research into what the ideal vacuum pressure is and which device will be best suited for using suction in RIRS without compromising safety and efficacy.(5)Potentially removing all fragments may also help avoid the need for a post-operative ureteral stent, unnecessary imaging, repeat procedures, and radiation time.(6)Since this is a simple technique using a scope channel as the conduit for aspiration, and works effectively with a 7.5 Fr scope, studies can determine if this can minimize the need and problems associated with the use of a UAS. This may decrease the need for presenting as well as increase first pass RIRS if the ureter is amenable to passing a scope.

## 5. Conclusions

In this study, we have described the new DISS technique which was successfully performed using two scope models on various stone locations and volumes. Our audit study was able to highlight its feasibility, utility, and limitations. The versatile nature of this technique needs an external multicenter validation and this will help to open up several future perspectives that can further improve efficacy, safety, and ergonomics in RIRS surgery.

## Figures and Tables

**Figure 1 jcm-11-05710-f001:**
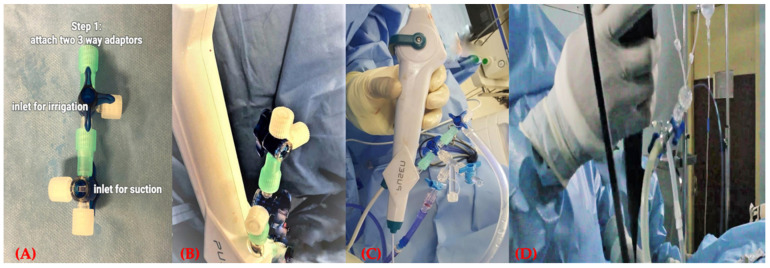
Assembling the DISS device. (**A**) Adaptor. (**B**,**C**) Adaptor attached to the scope. (**D**) Handling the scope with the adaptor attached.

**Figure 2 jcm-11-05710-f002:**
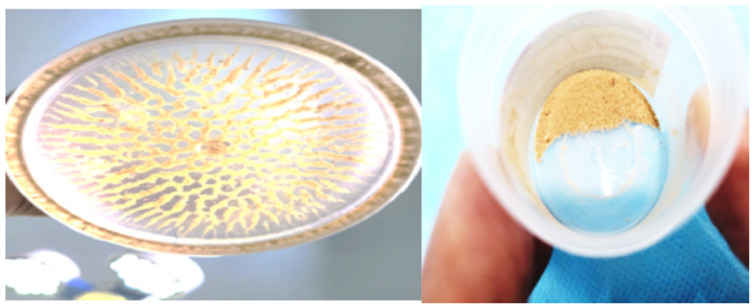
Dust aspirated through the DISS device.

**Table 1 jcm-11-05710-t001:** Patients’ characteristics. RIRS: retrograde intrarenal surgery. PCNL: percutaneous nephrolithotripsy. SWL: shock wave lithotripsy.

	DISS RIRS (n = 30)	SUAS RIRS (n = 28)	*p*-Value
Age, years, median (25th–75th percentiles)	46.0 (39.3–53.8)	49.0 (37.0–61.0)	0.360
Gender, n (%) Female Male	10 (33.3) 20 (66.7)	13 (46.4) 15 (53.6)	0.453
History of lithiasis, n (%) First time Recurrence	20 (66.7) 10 (33.3)	20 (71.4) 8 (28.6)	0.914
Haematuria only at presentation, n (%)	18 (60.0)	1 (3.56)	<0.001
Pain only at presentation, n (%)	12 (40.0)	6 (21.43)	<0.001
Haematuria and pain at presentation, n (%)	0	4 (14.28)	0.104
Fever at presentation, n (%)	0	5 (17.86)	0.051
Elevated creatinine at presentation, n (%)	1 (3.3)	5 (17.9)	0.167
Tamsulosin pre, n (%)	1 (3.3)	22 (78.6)	<0.001
Stone largest size, mm, median (25th–75th percentiles)	22.0 (18.0–28.8)	13.0 (11.8–15.0)	<0.001
Hounsfield unit, median (25th–75th percentiles)	1000 (544–1180)	1020 (1000–1070)	0.340
Multiple stones, n (%)	12 (40.0)	0	<0.001
Stone location, upper pole, n (%)	6 (20.0)	6 (21.44)	0.99
Stone location, middle pole, n (%)	24 (80.0)	11 (39.28)	<0.001
Stone location, lower pole, n (%)	12 (40.0)	11 (39.28)	0.99
Surgical time, minutes, median (25th–75th percentiles)	80.0 (60.0–100)	47.5 (41.5–60.3)	<0.001
Hospital stay, days, median (25th–75th percentiles)	1.00 (0.667–0.80)	1.00 (1.00–2.00)	0.002
Residual fragments, n (%) Single residual fragments, n (%) Multiple residual fragments, n (%)	10 (33.3) 2 (6.7) 8 (26.7)	10 (35.7) 5 (17.9) 5 (17.9)	0.99
Complications, n (%) Fever Haematuria Sepsis	11 (36.7) 0 0	7 (25.0) 2 (7.1) 0	0.4990.441
Intervention for residual fragments, n (%) Observation RIRS PCNL SWL	0 10 (33.3) 0 0	8 (28.6) 0 0 1 (3.6)	

## Data Availability

Data will be provided by the corresponding author upon a reasonable request.

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
