# Peer review of "Technique, Feasibility, Utility, Limitations, and Future Perspectives of a New Technique of Applying Direct In-Scope Suction to Improve Outcomes of Retrograde Intrarenal Surgery for Stones"

_jcm, 2022, doi:10.3390/jcm11195710_

Round 1

Reviewer 1 Report

The authors described a simple and cost-effective suction technique that can directly remove dust during laser lithotripsy via flexible ureteroscope. They called direct in scope suction (DISS) technique and I can see its potential effectiveness in practice. 

The content of the manuscript is worthwhile and the the existing literature has been reviewed adequately. But frankly, this study has some non-methodological issues that needed some revisions before it can be accepted. I have some specific comments that the authors must address:

 1. The indications of RIRS were controversial and the stone free rates were  

inconsistent. If available, I suggest the author add these in the Introduction section.

2. As a clinician, I am extremely concerned about the protocol of your management. For example, do you have routine pre-op or post-op CT. How about the protocols of follow-up? If available, I suggest the authors provide some details and your experience in the Discussion section.

3. In the DISS group, is there not necessary for the using of sheath or stone basket? Is it safe for the insertion and manipulation of flexible scope?

4. I think the author can highlight some issues that need to be addressed during follow-up period in the Discussion section.

Author Response

The authors described a simple and cost-effective suction technique that can directly remove dust during laser lithotripsy via flexible ureteroscope. They called direct in scope suction (DISS) technique and I can see its potential effectiveness in practice.

REPLY: We would like to thank you for your positive comment on our study

The content of the manuscript is worthwhile and the existing literature has been reviewed adequately. But frankly, this study has some non-methodological issues that needed some revisions before it can be accepted. I have some specific comments that the authors must address:

  1. The indications of RIRS were controversial and the stone free rates were inconsistent. If available, I suggest the author add these in the Introduction section.

REPLY. We would like to thank for suggesting us this important part. As suggested, we have modified the introduction as follows “Retrograde intrarenal surgery (RIRS) has gained popularity and is accepted in guidelines as a primary modality for the management of renal stones up to 2 cm [1]. With the advent of newer-generation flexible ureterorenoscopes, Holmium:YAG laser lithotripsy, and Thulium fiber, this trend is also being exploited in stones larger than 2 cm [2], pediatric cases [3], anomalous kidneys [4], and competing with percutaneous nephrolithotripsy (PCNL) [5]. However, despite the advances in technology challenges in achieving a high stone-free rate (SFR) in large stones greater than 2 cm with series reporting SFR between 56.7% (1) in single procedure  to 89.3% in a mean average of1.6  procedures (2) and in anomalous kidneys with SFR of 76.6% (3) to 79.2%,(4). Hence, much consideration needs to be given to choose RIRS over PCNL as the first choice of intervention.”

  1. As a clinician, I am extremely concerned about the protocol of your management. For example, do you have routine pre-op or post-op CT. How about the protocols of follow-up? If available, I suggest the authors provide some details and your experience in the Discussion section.

REPLY. We would like to thank for this important comment. We have now modified our protocol statements as you have suggested as follows “Postoperative follow-ups were performed between 4 and 6 weeks to evaluate complications and at 3 weeks to assess residual fragments (RF) which were evaluated with KUB and ultrasound or computed tomography scan if deemed necessary. Stone-free status was defined as the absence of a single residual fragment >3 mm or in the absence of multiple fragments of any size.”

We have also reported the RF and the subsequent intervention rates in table 1

Further, kindly note that we have stated this in our original discussion as well:

“Our audit study was that the post-operative imaging done as part of an early follow-up study (3 weeks) in the DISS group was either an x-ray or CT scan and all the 10  patients with RF who had only an X-ray were counseled for a repeat RIRS. However, all 28 patients who had  SUAS underwent a CT scan, and hence based on the RF location and size only 1 patient needed ESWL as an ancillary procedure.”

  1. In the DISS group, is there not necessary for the using of sheath or stone basket? Is it safe for the insertion and manipulation of flexible scope?

REPLY: We would like to thank you for this very valuable point which was missed in our paper. We have now added in advantages the following sentences “DISS technique could potentially minimize  usage of accessories like baskets for fragment removal and by suction aspiration of all the dust improve SFR. Yet, it is indeed safe to insert and manipulate scope. In fact, the no touch technique had been safety described as a standard of care by Grasso et al. We experienced no difficulty and only those patients in whom access was possible easily we proceeded to do surgery”.

  1. I think the author can highlight some issues that need to be addressed during follow-up period in the Discussion section.

REPLY. We would like to thank you for the suggestion but we have an entire section addressing the intra operative tricks and limitations. We think that this part should enough for follow-up

Reviewer 2 Report

Please explain that did you observe bleeding during the procedure in DISS group? Generally, aspiration procedue in collecting system may lead to bleeding. 

Author Response

Please explain that did you observe bleeding during the procedure in DISS group? Generally, aspiration procedure in collecting system may lead to bleeding.

REPLY. We would like to thank you for this very important comment. We apologies that we overlooked this in our discussion. We have now added  in the section technical consideration the following part:

Point 8:

Surgeons using suction either by DISS or SUAS must be aware that mild pelvic-caliceal system bleeding can be expected if too much suction pressure is applied which causes mild accidental mucosal petechial injury or inadvertent suctioning of the mucosa due to a collapsed system or due to rubbing of the sheath when it is moved across the ureteropelvic junction into pelvic-caliceal system to aspirate fragments. In our experience, this was not bothersome as it does not hamper vision, is transient and stops in a few seconds when suction is temporarily released and adequate irrigation is provided. However, caution must be exerted in being gentle with manipulation of scope or UAS in the pelvic-caliceal system.

Reviewer 3 Report

This is a retrospective study of 30 patients who underwent RIRS with a new technique of direct in scope suction (DISS). The authors compared this group with 28 patients who received RIRS with a suction ureteral access sheath (SUAS). Compared to the SUAS group, patients who received DISS had longer surgical time and shorter length of stay, yet no difference in terms of postoperative complications was noted between the two groups. Despite the similar incidence of residual fragments between the two groups, more patients in the DISS groups underwent re-RIRS. The authors concluded that RIRS with DISS technique was feasible with an acceptable rate of retreatment as compared to RIRS with SUAS.

Herein are my comments:

1.     The authors should explain the pros and cons of the DISS technique compared to the traditional techniques to justify their research question/goal.

2.     Study variables and outcomes (primary, secondary) should be mentioned in the “Methods”.

3.     What is the definition of “residual stone” in this study?

4.     How were the patients assigned to receive DISS vs SUAS?

5.     It seems that the DISS group included patients with more complex stones (e.g., larger, multiple, higher OR time). The authors should elaborate on this.

6.     The authors should add more details about the surgical technique, such as using access sheath or JJ placement, and postop care, including visit intervals, medications, and imaging.

7.     The authors reported the median (IQR) of hospital stay in DISS group as 1 (0.667-0.80). This should be wrong.

8.     What was the reason for a higher rate of re-intervention in the DISS group compared to SUAS, despite a similar rate of residual fragments?

9.     The authors reported only hematuria and fever as postop complications. Were there any other complications, e.g., UTI, LUTS, etc.? What was the time period for recording the complications: in-hospital, 30-day, or 90-day?

Author Response

This is a retrospective study of 30 patients who underwent RIRS with a new technique of direct in scope suction (DISS). The authors compared this group with 28 patients who received RIRS with a suction ureteral access sheath (SUAS). Compared to the SUAS group, patients who received DISS had longer surgical time and shorter length of stay, yet no difference in terms of postoperative complications was noted between the two groups. Despite the similar incidence of residual fragments between the two groups, more patients in the DISS groups underwent re-RIRS. The authors concluded that RIRS with DISS technique was feasible with an acceptable rate of retreatment as compared to RIRS with SUAS.

Herein are my comments:

  1. The authors should explain the pros and cons of the DISS technique compared to the traditional techniques to justify their research question/goal.

REPLY. We would like to thank you for this very important comment. Whilst we have in depth explained how RIRS outcomes may benefit by adding suction, as kindly recommended by you, to better  improve the readers understanding on the transition from traditional RIRS  to suction aided RIRS, we have now added to the discussion the following sentence “Traditionally  the focus in RIRS has been on scopes, lasers and utility of access sheaths, irrigation devices and ability to effectively yet safely perform RIRS and amongst ...."

We have also stated in discussion the following “DISS technique was formulated to overcome these very limitations.” We have now modified this line as “DISS technique was formulated to overcome these very limitations yet maintaining the principles of traditional RIRS where endourologists can continue to perform RIRS in the exact same way as has been traditionally done with a simple add-on modification as described in the surgical technique to perform DISS.”

  1. Study variables and outcomes (primary, secondary) should be mentioned in the “Methods”.

REPLY. We would like to thank for this suggestion. We do apologizes for missing this in our original manuscript. We have now added the following sentences in Materials and Methods “Primary outcome was difference in SFR between the two techniques. Secondary outcome was difference in complications.”

  1. What is the definition of “residual stone” in this study?

REPLY. We would like to thank for this important comment. We have now modified our protocol statements as you have suggested as follows “Postoperative follow-up was performed at 3 weeks to evaluate complications and assess residual fragments which were evaluated with KUB and ultrasound or computed tomography scan if deemed necessary. Stone-free status was defined as the absence of a single residual fragment >3 mm or in the absence of multiple fragments of any size.”

  1. How were the patients assigned to receive DISS vs SUAS?

REPLY. We would like to thank for this important comment. This was a retrospective and comparative study. As it was an audit study patients were assigned to receive either based on the surgeons personal preference.

We have added  this point to the section Audit study: “Patients received either of the procedure based on surgeons own preference and were included only if ureteric access was feasible. Patients in whom a ureteric access for RIRS was impossible on the first attempt were then pre-stented and  excluded from this study”

  1. It seems that the DISS group included patients with more complex stones (e.g., larger, multiple, higher OR time). The authors should elaborate on this.

REPLY. We would like to thank for this important comment. We agree with you but as mentioned, this was incidentally noted .As it was a retrospective study we present the findings as they were recorded.

  1. The authors should add more details about the surgical technique, such as using access sheath or JJ placement, and postop care, including visit intervals, medications, and imaging.

REPLY. We would like to thank you for this important comment.

  • Regarding surgical technique: A whole subsection 2.2 highlights this.
  • Regarding utility of UAS: As mentioned all DISS patients had no UAS and other group had 11/ 13 fr Suction UAS from Clearpetra company.
  • Regarding the utility of JJ placement: As you had kindly highlighted we have now explained as in point 4 how patients were chosen and none were pre-stented. We have now added as per your kind recommendation in surgical technique. “Post procedure, all patients had a double J stent placed as standard of care in both groups.”
  • Regarding imaging, residual fragments were evaluated with KUB and ultrasound or computed tomography scan if deemed necessary (this was added in the text)
  1. The authors reported the median (IQR) of hospital stay in DISS group as 1 (0.667-0.80). This should be wrong.

REPLY. We would like to thank you for this comment. data are correct and presented as median and 25th-75th percentiles and not as IQR

  1. What was the reason for a higher rate of re-intervention in the DISS group compared to SUAS, despite a similar rate of residual fragments?

REPLY. We would like to thank you for this important comment. As DISS is a new technique, it was the surgeons decision after discussion with patients.  However, we would like to point out that while not intentional all DISS group all 10 patients with RF had RIRS as shown in table. These patients had only a XRay and not a CT scan for follow up. This might have erroneously contributed to overtreatment

Further, because of the concerns that arise when a new procedure is designed and performed in order to avoid any major complications due to RF the surgeons chose reintervention in this group.

  1. The authors reported only hematuria and fever as postop complications. Were there any other complications, e.g., UTI, LUTS, etc.? What was the time period for recording the complications: in-hospital, 30-day, or 90-day?

REPLY. We would like to thank you for this important comment. As routine folIow-up, we do not evaluate post-operative UTI's unless patients have fever or sepsis. LUTS as a sequel of performing RIRS is not evaluated as routine hence that information is not available for discussion . We do apologies for that inconvenience.

Round 2

Reviewer 1 Report

The authors well addressed all of my comments. Although the manuscript presents some limitations, the authors have clearly stated the limits of their study and improved the description of doubtful aspects. For my point of view the manuscript can be accepted for publication, however, the authors have to address also the other reviewers' comment and wait for their final approval.